# Cue-inhibited ventrolateral periaqueductal gray neurons signal fear output and threat probability in male rats

Kristina M Wright[1]*, Thomas C Jhou[2], Daniel Pimpinelli[1], Michael A McDannald[1]*

[1]Psychology Department, Boston College, Chestnut Hill, United States; [2]Department of Neuroscience, Medical University of South Carolina, Charleston, United States

**Abstract** The ventrolateral periaqueductal gray (vlPAG) is proposed to mediate fear responses to imminent danger. Previously we reported that vlPAG neurons showing short-latency increases in firing to a danger cue – the presumed neural substrate for fear output – signal threat probability in male rats (Wright et al., 2019). Here, we scrutinize the activity vlPAG neurons that *decrease* firing to danger. One cue-inhibited population flipped danger activity from early inhibition to late excitation: a poor neural substrate for fear output, but a better substrate for threat timing. A second population showed differential firing with greatest inhibition to danger, less to uncertainty and no inhibition to safety. The pattern of differential firing reflected the pattern of fear output, and was observed throughout cue presentation. The results reveal an expected vlPAG signal for fear output in an unexpected, cue-inhibited population.

DOI: https://doi.org/10.7554/eLife.50054.001

*For correspondence:
wrightko@bc.edu (KMW);
michael.mcdannald@bc.edu
(MAM)

Competing interests: The authors declare that no competing interests exist.

## Introduction

The ventrolateral periaqueductal gray (vlPAG) is an essential node in a neural circuit for defensive behavior. In the prevailing view of the defensive circuit, threat estimates originate in the amygdala and are relayed to the vlPAG to organize the behavioral components of fear output, such as freezing (*Fanselow, 1994*). A multitude of studies have observed a population of vlPAG single-units showing short-latency excitation to cues predicting foot shock (*Tovote et al., 2016*; *Watson et al., 2016*; *Ozawa et al., 2017*; *Groessl et al., 2018*). While short-latency, cue-excitation would provide a suitable neural substrate for freezing output, cue-excited neural activity did not consistently track freezing in each of these studies. Recently, we recorded vlPAG single-unit activity while rats underwent fear discrimination in which three auditory cues predicted unique foot shock probabilities: danger (p = 1.00), uncertainty (p = 0.375) and safety (p = 0.00) (*Wright and McDannald, 2019*). As in previous studies, we identified single-units with short-latency excitation to cue onset. Somewhat surprisingly, onset single-unit activity reflected the foot shock probability associated with each cue, rather than the level of fear demonstrated to that cue. Thus, short-latency excitatory responses reflect information about threat probability.

Although cue-excited single-units have been the focus of a neural substrate for fear output, cue-inhibited vlPAG single-units have also been found (*Tovote et al., 2016*). Further, optogenetic inhibition of this functional type promotes freezing. Among the vlPAG single-units recorded in our previous study (*Wright and McDannald, 2019*), a considerable number inhibited activity on cue presentation (91/245,~37% of single-units recorded), particularly to danger. The goal of the current study was to scrutinize these cue-inhibited, vlPAG single-units and determine whether they may provide a suitable neural substrate for fear output.

# Results

Rats were trained to nose poke in a central port in order to receive a food pellet from a cup below. During fear discrimination, three distinct auditory cues predicted unique foot shock probabilities: danger (p = 1.00), uncertainty (p = 0.375) and safety (p = 0.00) (*Figure 1A*). Trial order was randomized each session. Fear was measured using a suppression ratio and was calculated by comparing nose poke rates during baseline and cue periods (*Pickens et al., 2009*; *Berg et al., 2014*; *Wright et al., 2015*; *DiLeo et al., 2016*; *Walker et al., 2018*). After eight discrimination sessions, rats were implanted with 16-wire, drivable microelectrode bundles dorsal to the vlPAG (*Figure 1B*). Following recovery, rats were returned to fear discrimination and activity was recorded. Single-units were isolated and held for the duration of each recording session. The electrode bundle was advanced ~40–80 μm between sessions to record from new single-units in subsequent sessions.

Rats showed excellent discriminative fear: high to danger, intermediate to uncertainty, and low to safety (*Figure 1C*). ANOVA for suppression ratio for the entire 10 s cue using trial-type as a factor (danger, uncertainty and safety) revealed a main effect of trial-type ($F_{2,174}$ = 592.00, p = $2.32 \times 10^{-78}$, $\eta_p^2$ = 0.87, observed power (op) = 1.00). We constructed 95% bootstrap confidence intervals for differential suppression ratios to determine if discrimination was observed between each cue pair. Indicative of full cue discrimination, the 95% bootstrap confidence interval did not contain zero for danger vs. uncertainty (Mean = 0.30, 95% CI [(lower bound) 0.24, (upper bound) 0.34]) and uncertainty vs. safety (M = 0.50, 95% CI [0.44, 0.56]) (*Figure 1C*).

## vlPAG neurons flip to excitation or sustain inhibition over cue presentation

We recorded 245 neurons in six male Long Evans rats over 88 fear discrimination sessions. We identified 91 neurons (~37% of all neurons recorded) showing significant decreases in firing rate to danger or uncertainty. Visualization of all cue-inhibited neurons revealed heterogeneous inhibition of danger firing during late cue presentation (*Figure 1D*). To determine whether this heterogeneity reflected the activity of two separate populations, we performed k-means clustering for all cue-inhibited neurons on early (first 5 s) and late (last 5 s) firing to danger. The first cluster (n = 45) consisted of neurons that were danger-inhibited early, but danger-excited late. These neurons are referred to as the Flip population. The second cluster (n = 46) consisted of neurons that were danger-inhibited early and late, and are referred to as the Sustain population. Independent samples t-tests for waveform properties revealed no differences between Flip and Sustain neurons, indicating these populations could only be distinguished by their function (*Figure 1E–G*): baseline firing, $t_{89}$ = 0.95, p = 0.343; half duration, $t_{89}$ = 0.77, p = 0.444; amplitude ratio $t_{89}$ = 0.10, p = 0.918.

## Flip and sustain populations show differential cue firing

A vlPAG signal for fear output should begin at cue onset, continue throughout cue presentation, and discriminate danger, uncertainty and safety. To determine if cue-inhibited vlPAG neurons complied with these requirements, we examined mean population activity over cue presentation for Flip and Sustain neurons. Flip neurons were initially inhibited to uncertainty and danger, but lesser to safety (*Figure 2A*). As cue presentation continued, inhibition to uncertainty weakened toward safety and firing to danger switched to excitation (*Figure 2A*). ANOVA for normalized firing rate (Z-score) for the 45 Flip neurons [*Figure 2A*; factors: trial-type (danger, uncertainty and safety) and bin (250 ms bins encompassing: 2 s baseline, 10 s cue, and 2 s delay)] revealed main effects of cue ($F_{2,88}$ = 16.58, p = $7.74 \times 10^{-7}$, $\eta_p^2$ = 0.27, op = 1.00) and bin ($F_{55,2420}$ = 14.83, p = $1.03 \times 10^{-114}$, $\eta_p^2$ = 0.25, op = 1.00), but most critically a cue x bin interaction ($F_{110,4840}$ = 7.85, p = $8.89 \times 10^{-106}$, $\eta_p^2$ = 0.15, op = 1.00). The population pattern was consistent across individual trials, though late danger excitation was least on the first and last trials (*Figure 2—figure supplement 1A–D*).

We constructed a 95% bootstrap confidence interval to determine if differential firing was observed early and late in cue presentation for Flip neurons. Differential firing was not observed to danger vs. uncertainty early (M = 0.04, 95% CI [−0.05, 0.11]), but was observed late when danger flipped to excitation (M = 0.35, 95% CI [0.26, 0.42]) (*Figure 2B*, plus signs). In contrast, differential firing was observed to uncertainty vs. safety early (M = −0.18, 95% CI [−0.26,−0.11]), but not late

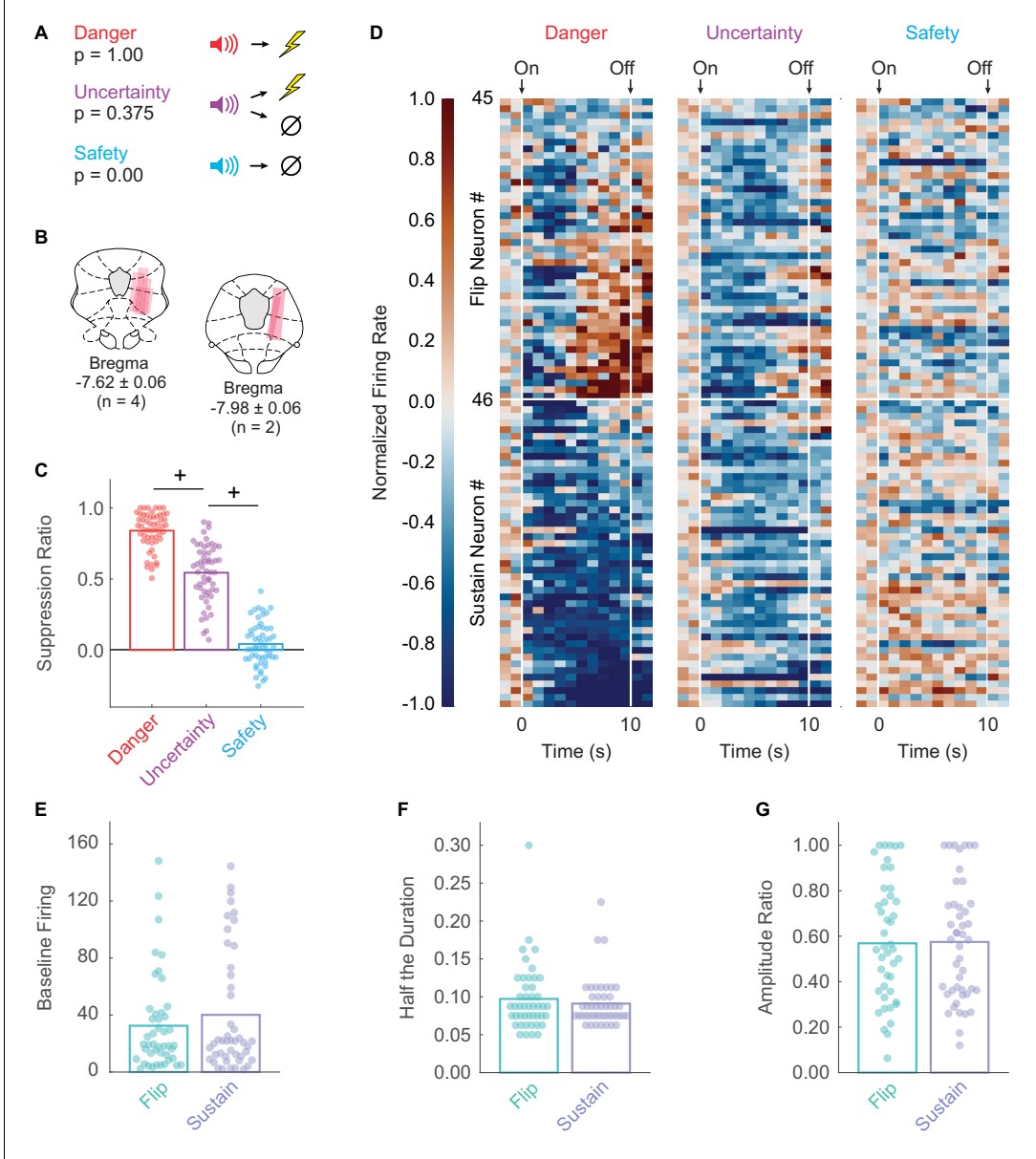

**Figure 1.** Fear discrimination, histology, heat plot and waveform characteristics. (**A**) Pavlovian fear discrimination consisted of three cues predicting unique foot shock probabilities: danger, p = 1.00 (red); uncertainty, p = 0.375 (purple); and safety, p = 0.00 (blue). (**B**) Microelectrode bundle placements for all rats (n = 6) and all neurons (n = 245) during recording sessions are represented by salmon bars. (**C**) Mean + individual (data points) suppression ratio for danger, uncertainty, and safety is shown for all recording sessions (n = 88). (**D**) Normalized firing rate in 1 s intervals is shown for each Flip (n = 45, top) and Sustain (n = 46, bottom) neuron for each trial type (danger, uncertainty and safety). Color scale for normalized firing rate is shown to the right; red indicates high firing and blue low firing. Cue onset and offset are indicated. Single-unit waveform properties of Flip (sea foam) and Sustain (periwinkle) neurons are shown: (**E**) baseline firing rate, (**F**) half the duration, and (**G**) amplitude ratio. +95% bootstrap confidence interval for differential suppression ratio does not contain zero.

DOI: https://doi.org/10.7554/eLife.50054.002

(M = −0.05, 95% CI [−0.14, 0.05]) (*Figure 2B*). Furthermore, the 95% bootstrap confidence interval for normalized firing did not contain zero for any cues in either period (*Figure 1B*, pound signs), indicating the Flip population was responsive to all cues.

Sustain neurons showed cue-selective inhibition of firing: danger < uncertainty < safety ≈ 0. This firing pattern was observed throughout cue presentation (*Figure 2C*). ANOVA for normalized firing rate [*Figure 2C*; factors maintained from above] revealed main effects of cue ($F_{2,86}$ = 72.25,

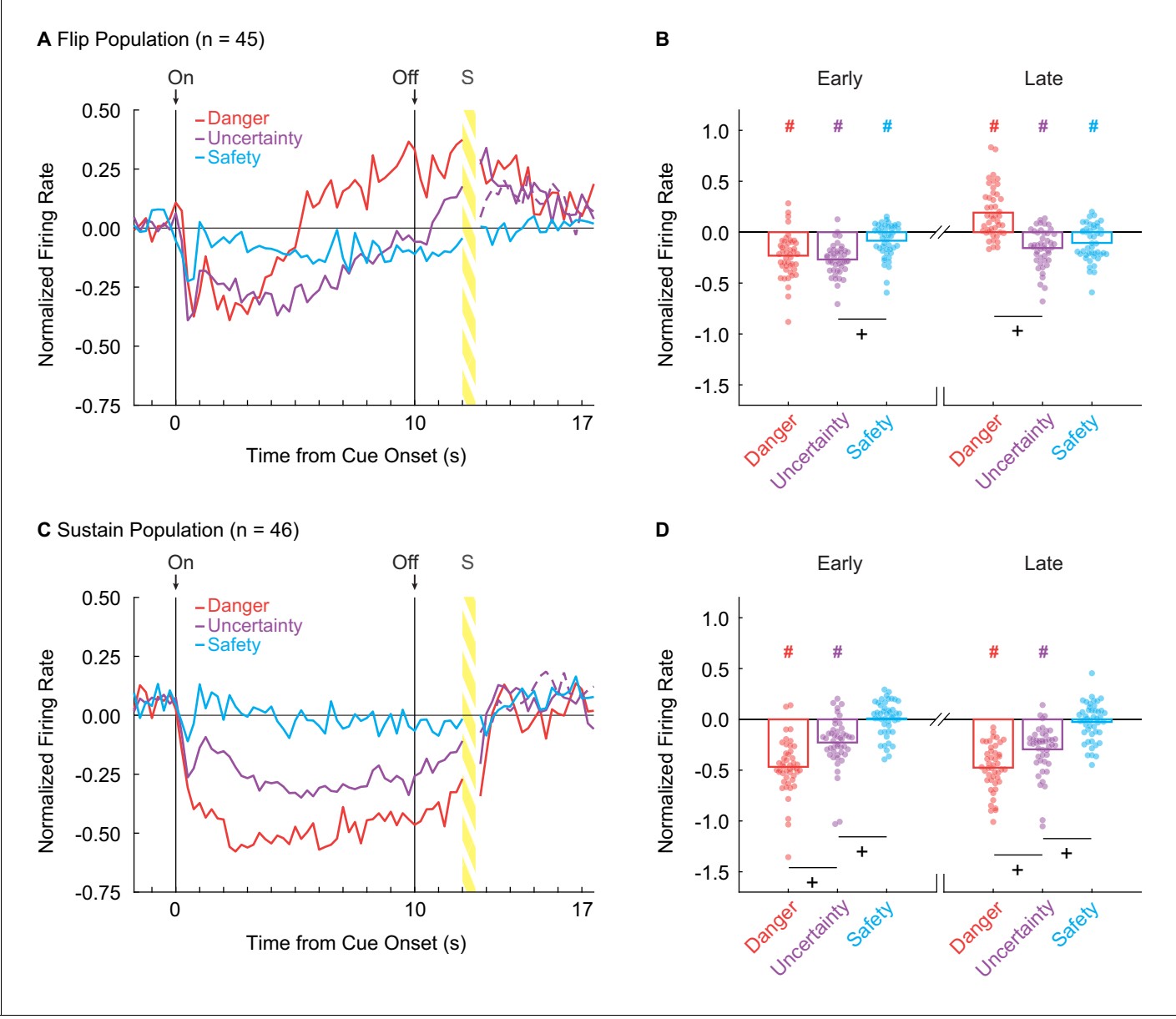

**Figure 2.** vlPAG neurons flip to excitation or sustain inhibition over cue presentation. (**A**) Mean normalized firing to danger (red), uncertainty (purple) and safety (blue) is shown for the 2 s pre-cue period, 10 s cue period, and 2 s delay period for the Flip population (n = 45). Cue onset (On) and offset (Off) are indicated by vertical black lines. (**B**) Mean (bar) and individual (data points), normalized firing for Flip neurons during the first 5 s of cue presentation (Early, left) and the last 5 s of cue presentation (Late, right) is shown for each cue. (**C**) Mean normalized firing for the Sustain population (n = 45), shown as in A. (**D**) Mean and individual (data points), normalized firing for Sustain neurons, as in B. [+]95% bootstrap confidence interval for differential firing does not contain zero. [#]95% bootstrap confidence interval for normalized firing does not contain zero.

DOI: https://doi.org/10.7554/eLife.50054.003

The following figure supplement is available for figure 2:

**Figure supplement 1.** Trial by trial firing for Flip and Sustain populations.

DOI: https://doi.org/10.7554/eLife.50054.004

p = $3.88 \times 10^{-19}$, $\eta_p^2$ = 0.63, op = 1.00) and bin ($F_{55,2365}$ = 14.91, p = $6.13 \times 10^{-115}$, $\eta_p^2$ = 0.26, op = 1.00), as well as a cue x bin interaction ($F_{110,4730}$ = 5.24, p = $1.17 \times 10^{-59}$, $\eta_p^2$ = 0.11, op = 1.00). The population pattern was consistent across all trials (*Figure 2—figure supplement 1E–H*). Selective firing was observed early and late in cue presentation. In support, the 95% bootstrap confidence interval for differential firing did not contain zero for danger vs. uncertainty (Early:

M = −0.24, 95% CI [−0.35,–0.14], Late: M = −0.18, 95% CI [−0.25,–0.11]) and uncertainty vs. safety (Early: M = −0.23, 95% CI [−0.32,–0.15], Late: M = −0.27, 95% CI [−0.35, 0.18]) (*Figure 2D*, plus signs). Even more, the 95% bootstrap confidence interval for normalized firing did not contain zero for danger and uncertainty during both periods, but *did* contain zero for safety during both periods (*Figure 1D*, pound signs). Not only was differential firing observed, but Sustain neurons were selectively responsive to danger and uncertainty.

## Population biases are evident in single-units

If the vlPAG signals fear output, one would expect population-level signals to be observed at the single-unit level. To examine this, we used sign tests to identify whether single-unit firing was biased away from zero during early and late cue presentation. Flip single-units were biased towards decreased firing to danger [Early: (p(sign) = $9.33 \times 10^{-9}$)] and uncertainty [Early: (p(s) = $5.89 \times 10^{-11}$)] during early cue presentation. Strikingly, and consistent with the population response, Flip neurons were biased towards *increased* firing to danger [Late: (p(s) = $8.24 \times 10^{-4}$)], but decreased firing to uncertainty [Late: (p(s) = $2.47 \times 10^{-4}$)] during late cue presentation. Contrary to the population result, there was no bias away from zero in single-unit firing to safety early or late. Single-unit biases of Sustain neurons mirrored those observed in the population. Sustain single-units showed a consistent bias toward decreased firing to danger [Early: p(s) = $3.08 \times 10^{-11}$, Late: p(s) = $2.84 \times 10^{-14}$)] and uncertainty [(Early: p(s) = $3.10 \times 10^{-7}$, Late: (p(s) = $5.10 \times 10^{-9}$)] throughout cue presentation. Further, Sustain single-units showed no bias in firing to safety in either cue period. Observing fully differential firing by single-units throughout cue presentation further marks Sustain neurons as a candidate for signaling fear output.

## Flip neurons switch threat probability signaling from early to late cue presentation

Descriptive analyses reveal two cue-inhibited populations with distinct temporal activity patterns. However, these analyses do not reveal the information signaled by each population. We used linear regression for single-unit firing to formally test the degree to which Flip and Sustain neurons signaled fear output and threat probability (*Figure 3*). For each single-unit, we calculated the normalized firing rate for each trial (32 total: six danger, six uncertainty shock, 10 uncertainty omission, and 10 safety) in 1 s bins over the course of cue presentation (14 s total: 2 s pre-cue, 10 s cue, 2 s post-cue). Fear output was the suppression ratio on that trial. Threat probability was the shock probability associated with the cue: danger: 1.00, uncertainty: 0.375 and safety: 0.00. Regression output for each single-unit was a beta coefficient quantifying the strength (|>0| = stronger) and direction (>0 = positive and <0 = negative) of the predictive relationship between the regressor and single-unit firing. Beta coefficients for single-units were subjected to ANOVA with regressor (fear output vs. threat probability) and interval (1 s cue intervals) as factors.

Single-unit regression revealed an early-to-late switch in threat probability signaling by Flip neurons (*Figure 3A*). ANOVA for beta coefficients with factors of regressor (fear output vs. threat probability) and interval was performed for three periods: baseline (two intervals), cue (10 intervals) and delay (two intervals). The baseline and delay ANOVAs returned no main effects or interaction (all F < 0.6, all p > 0.4). In contrast, the cue ANOVA found significant main effects, but most critically a regressor x interval interaction ($F_{9,396}$ = 3.56, p = $2.85 \times 10^{-4}$, $\eta_p^2$ = 0.075, op = 0.990). The interaction was driven by negative beta coefficients for fear output and threat probability in two, early cue intervals (95% bootstrap confidence interval did not contain zero, pound signs), that gave way to positive beta coefficients specific to threat probability in all late cue intervals (95% bootstrap confidence interval did not contain zero, pound signs; *Figure 3A*). Further supporting the interaction, beta coefficients for Flip single-units were not biased away from zero for fear output and threat probability during the first 5 s cue period [Probability Early: p(s) = 0.37, Fear Output Early: p(s) = 0.14] (*Figure 3B*). However, there was positive bias toward threat probability, but not fear output, during the last 5 s cue period [Probability Late: p(s) = $2.47 \times 10^{-4}$, Fear Output Late: p(s) = 0.77] (*Figure 3C*). Fear responses are sustained for the cue duration, yet Flip neurons do not consistently signal threat probability or fear output in early cue presentation. The inconsistency in signaling reveals that Flip neurons are not a suitable neural substrate for governing fear output throughout cue presentation.

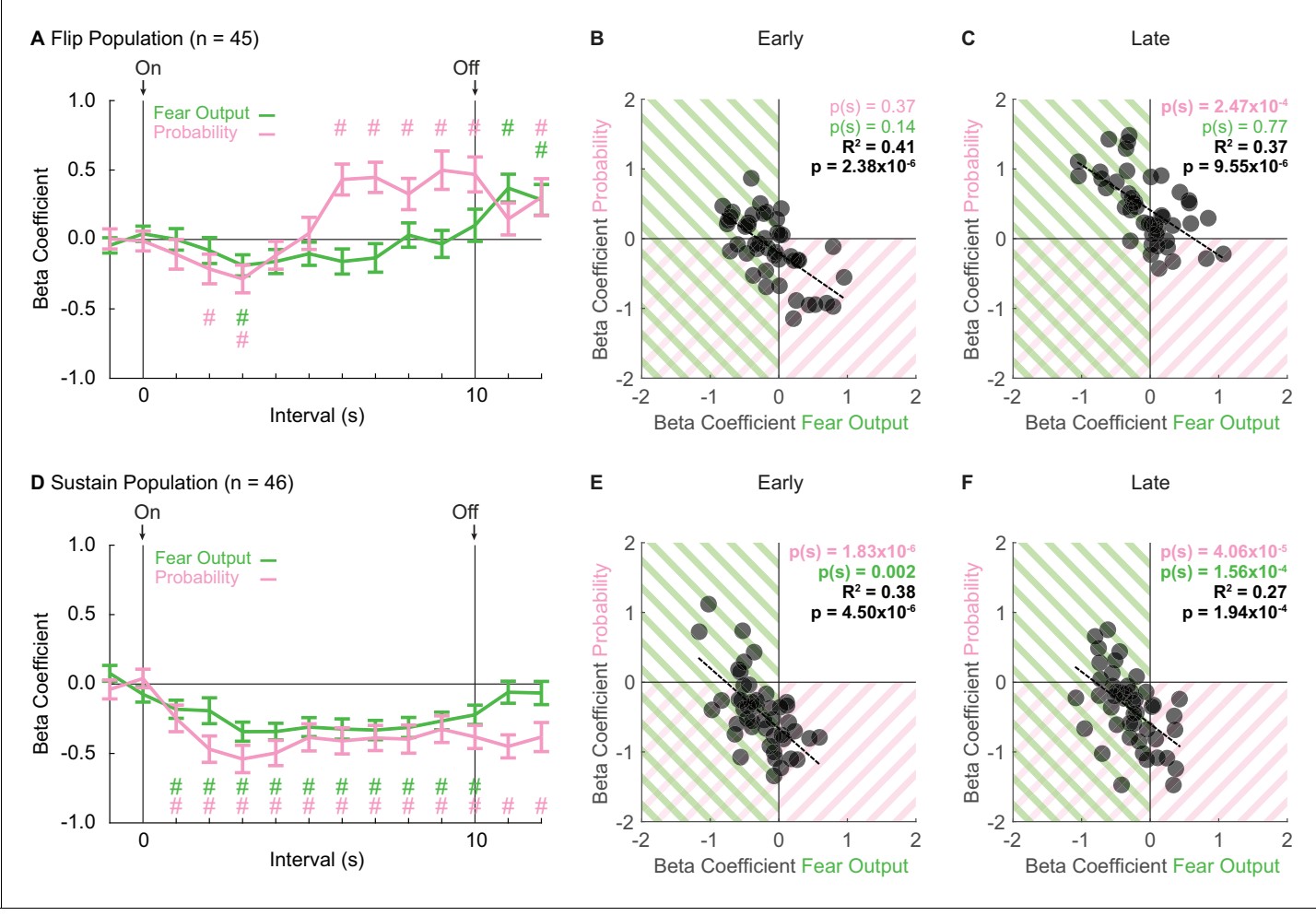

**Figure 3.** Sustain and Flip populations signal threat probability and fear output. (**A**) Mean ± SEM beta coefficients are shown for each regressor (probability: pink, fear output: green), in 1 s intervals, for the Flip population (n = 46). (**B**) Beta coefficients during the first, 5 s of cue presentation (Early) for fear output and threat probability are plotted for all Flip neurons. Black dashed trendline, the square of Pearson's correlation coefficient ($R^2$) with associated p value, and sign test p value demonstrating regressor bias shown. Background shading indicates negative beta coefficients, color coded by regressor. (**C**) Beta coefficients during the last, 5 s of cue presentation (Late) for fear output and threat probability are plotted for all Flip neurons. (**D**) Mean ± SEM beta coefficients are shown for each regressor for the Sustain population (n = 46). All graph properties maintained from A. (**E**) Beta coefficients during Early cue presentation for fear output and threat probability are plotted for all Sustain neurons. All graph properties maintained from B. (**F**) Beta coefficients during Late cue presentation for fear output and threat probability are plotted for all Sustain neurons. #95% bootstrap confidence interval for beta coefficient does not contain zero.

DOI: https://doi.org/10.7554/eLife.50054.005

## Sustain neurons signal fear output and threat probability throughout cue presentation

Linear regression revealed consistent signals for fear output and threat probability in Sustain neurons. Beta coefficients were negative at cue onset for each regressor, and maintained this negativity throughout cue presentation (*Figure 3D*). ANOVA for beta coefficients with factors of regressor and interval was performed as before for baseline, cue and delay. The baseline ANOVA returned no main effects or interaction (all F < 1, all p > 0.3). The cue ANOVA found only a main effect of bin ($F_{9,405}$ = 4.23, p = 2.90×10$^{-5}$, $\eta_p^2$ = 0.086, op = 0.997), indicating similar signaling of fear output and threat probability. The delay ANOVA found only a main effect of regressor ($F_{1,45}$ = 7.27, p = 0.01, $\eta_p^2$ = 0.14, op = 0.751), indicating a difference in signaling of fear output and threat probability during the delay period. For each regressor over the 10, 1 s cue intervals, the 95% bootstrap confidence interval did not contain zero, indicating that fear output and threat probability signaling

were observed throughout cue presentation. Consistent with equivalent signaling of fear output and probability throughout cue presentation, single-unit beta coefficients for each regressor were biased away from zero for fear output and threat probability during early and late cue presentation [Probability (Early: p(s) = $1.83 \times 10^{-6}$ and Late: p(s) = $4.06 \times 10^{-5}$), Fear Output (Early: p(s) = 0.002 and Late: p(s) = $1.56 \times 10^{-4}$)] (*Figure 3E & F*). Further, single-unit beta coefficients for threat probability and fear output were correlated early and late (Early: $R^2$ = 0.41, p = $2.38 \times 10^{-6}$ and Late: $R^2$ = 0.37, p = $9.55 \times 10^{-6}$). The majority of Sustain single-units showed negative beta coefficients for both regressors. However, even the extremes of the distribution showed signaling for both regressors, albeit in opposing directions. Sustain neurons signal fear output *and* threat probability throughout cue presentation.

## Differential threat tuning in flip and sustain neurons

The threat probability regressor in the above analyses utilized the actual shock probability assigned to uncertainty (0.375). Of course, the subjects, and by extension their neurons, had no a priori knowledge of the actual shock probability. Thus, it is possible that Flip and Sustain single-units are 'tuned' to alternative shock probabilities. To test this, we performed single-unit linear regression for normalized firing in each 1 s interval as before, maintaining the probabilities for danger (1.00) and safety (0.00), but incrementing the probability assigned to uncertainty from 0 to 1 in 0.125 steps (0.000, 0.125, 0.250, 0.375, 0.500, 0.625, 0.750, 0.875, and 1.000). Threat probability beta coefficients were averaged over early and late cue presentation. The mean beta coefficient for each uncertainty assignment is plotted as a threat-tuning curve, early and late, for each population (*Figure 4*).

Flip neurons were tuned to alternative foot shock probabilities and this tuning changed from early to late cue presentation. Early threat overestimation (equating uncertainty to danger) gave way to late underestimation (equating uncertainty to safety). The tuning curve trough for early cue presentation occurred at 0.750 (*Figure 4A*, light pink); overshooting the actual probability of 0.375 (*Figure 4A*, dashed black line) and exceeding mean fear output (*Figure 4A*, dashed green line). The tuning curve peak for late cue presentation occurred at 0.250 (*Figure 4A*, dark pink); undershooting the actual probability (*Figure 4A*, dashed black line). ANOVA for beta coefficient with factors of time (early vs. late) and uncertainty assignment (9) found both main effects and the interaction to be significant (all F > 13, all p < 0.001). The relative firing patterns of Flip neurons do not approximate the actual probability of foot shock or the pattern of fear output, and further indicate that these neurons are unlikely to govern fear output.

In contrast, Sustain neuron tuning consistently fell between the bounds of the actual foot shock probability and the mean fear output, changing only subtly over cue presentation (*Figure 4B*). The trough of the tuning curve occurred at an assignment of 0.500 early and at an assignment of 0.625 late. ANOVA found a main effect of assignment ($F_{8,360}$ = 5.66, p = $8.85 \times 10^{-7}$, $\eta_p^2$ = 0.112, op = 1.00) and a time x assignment interaction ($F_{8,360}$ = 3.25, p = 0.001, $\eta_p^2$ = 0.067, op = 0.971). The stability of Sustain tuning and the bias toward mean fear output further suggests this population as a candidate for fear output.

## Discussion

We set out to scrutinize cue-inhibited vlPAG neurons to determine if their activity reflected fear output. Unexpectedly, we found one population inhibited to danger early, but excited to danger late. Although Flip neurons are not suitable candidates for signaling fear output, they may provide information about the anticipated time of foot shock. Consistent with this speculation, peak activity of Flip neurons occurred just prior to shock presentation and declined toward baseline shortly after. This finding is in general accord with studies showing that the shift from distal to proximate threats corresponds with a shift from prefrontal to periaqueductal activity (*Mobbs et al., 2007*; *Mobbs et al., 2010*). Further, the patterned activity of Flip neurons is similar to our previously reported 'Ramping' population, which showed little change in activity at cue presentation but increased firing toward shock delivery (*Wright and McDannald, 2019*). Flip and Ramping neurons may comprise a single, functional population, and suggest a central role for the vlPAG in threat timing.

The patterned activity of Sustain neurons complies with basic assumptions of a neural correlate for fear output. Sustain neurons decreased firing to threat-related cues, but did not decrease firing

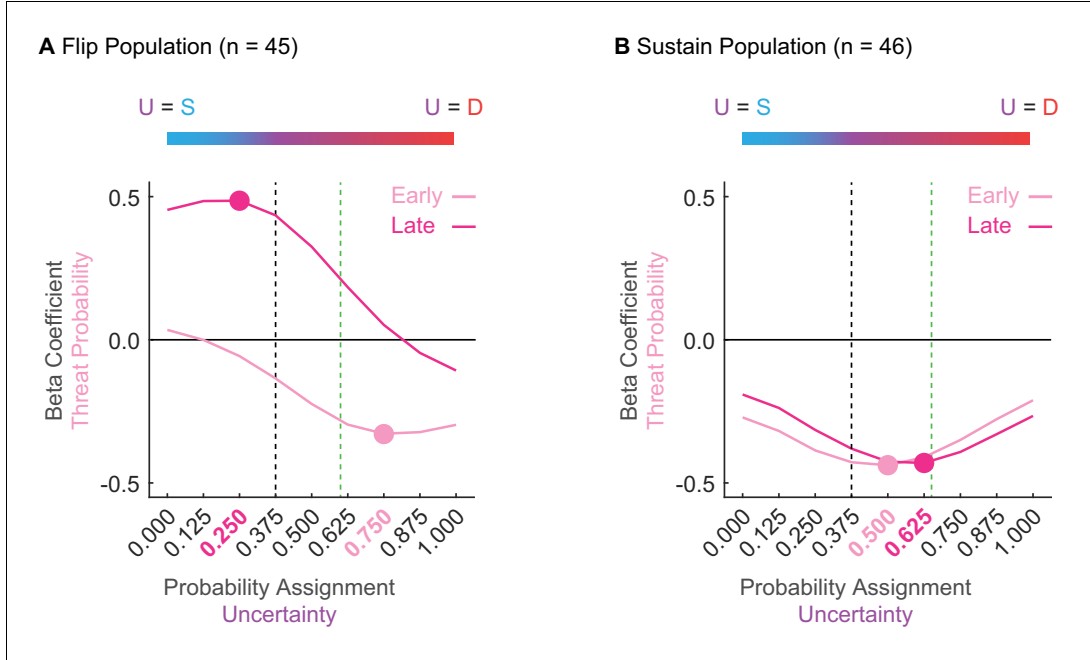

**Figure 4.** Sustain and Flip populations probability tuning. (**A**) Mean beta coefficient for probability is shown for early (light pink) or late (hot pink) cue presentation for each of nine uncertainty assignments for the Flip population (n = 45). The peak or trough of each curve is indicated by a single point with the corresponding uncertainty assignment highlighted in the same color on the x axis below. Black dashed line indicates the actual foot shock probability associated with uncertainty (p = 0.375). Green dashed line indicates the mean proportional distance of uncertainty between danger and safety (suppression ratio). The blue-to-red color bar at the top of the figure demonstrates that a leftward shift along the x-axis reflects an uncertainty assignment similar in quality to safety (p = 0.000) versus those that would be more similar to a danger cue on the far right (p = 1.000). (**B**) All graph properties maintained from A, but applied to the Sustain population.

DOI: https://doi.org/10.7554/eLife.50054.006

to safety. Neural activity fully discriminated between danger, uncertainty and safety from cue onset through shock presentation and returned to baseline shortly thereafter. These population biases were observed in single-units. Interestingly, while threat probability signaling was observed in Sustain single-units, the probability to which neurons were tuned exceeded the actual probability of 0.375 and better approximated fear output. Although we do not have causal evidence that Sustain neurons drive a discriminative fear response, we have identified a complete neural correlate for fear output in a population of vlPAG neurons.

An influential theory posits that vlPAG output is achieved through a disinhibition mechanism (*Tovote et al., 2016*). GABA interneurons (with high baseline firing rates) receive inputs from GABAergic neurons in the central amygdala. Central amygdala GABA neurons increase firing to danger cues, inhibiting and reducing firing of vlPAG interneurons. This stops the local inhibition of glutamate neurons that now increase firing to promote freezing through downstream projections. Consistent with a disinhibition mechanism, we observed Sustain neurons that have high baseline firing rates (*Figure 1A*, right). However, we observed many Sustain neurons that had low baseline firing rates, including those with baseline rates just above zero. While we cannot conclusively determine neuron type from baseline firing, it is likely that cue-inhibited neurons are not uniformly GABAergic interneurons. While inconsistent with a pure disinhibition mechanism, our results are consistent with an alternative view in which the vlPAG contains unique output populations that separately convey information via excitation and inhibition (*Lau and Vaughan, 2014*). Most likely, the vlPAG utilizes disinhibition, as well as independent signaling via cue-excited and cue-inhibited projection populations.

Altogether, our previous and present findings reveal diverse temporal responding and threat signaling in the vlPAG. Our previous study (*Wright and McDannald, 2019*) found a cue-excited

population that phasically fired at cue onset and exclusively signaled threat probability. That study also identified cue-excited population of 'Ramping' neurons that increased firing over cue presentation, and prioritized threat probability signaling over fear output (*Wright and McDannald, 2019*). The present findings expand the functional diversity of vlPAG neurons. Patterned activity and signaling of Flip neurons resembles that of our previously identified 'Ramping' neurons, perhaps marking a single functional class critical for timing impending noxious events. Patterned activity and signaling of Sustain neurons was more consistent with fear output, but also contained information about threat probability. Observing robust vlPAG threat-related activity is expected, given its essential role in defensive behavior (*Bandler and Depaulis, 1991*; *Fanselow, 1991*; *Carrive et al., 1997*). However, the diversity of information contained in these signals is surprising. Concurrent with our findings, there is increasing evidence that vlPAG dysfunction may contribute to a variety of psychiatric disorders (*George et al., 2019*). Understanding the factors that determine vlPAG neuron function: cell-type (*Li et al., 2016*), transcriptome (*Okaty et al., 2015*; *Okaty et al., 2019*), connectome (*Rozeske et al., 2018*) and more (*McPherson et al., 2018*), will be essential to understanding the neural mechanisms underlying adaptive and maladaptive threat behavior.

# Materials and methods

**Key resources table**

| Reagent type (species) or resource | Designation | Source or Reference | Identifier | Additional information |
|---|---|---|---|---|
| Antibody | Anti-Tryptophan Hydroxylase (sheep polyclonal) | Sigma | Cat # T8575 RRID:AB_1080792 | [1:1000] in 0.05M PBS |
| Antibody | Biotinylated Anti-Sheep (rabbit clonality unknown) | Vector Labs | Cat # PK-6106 RRID:AB_2336217 | [1:200] in 0.05M PBS |
| Chemical Compound, Drug | Normal Rabbit Serum | Vector Labs | Cat # PK-6106 RRID:AB_2336825 | 1% in 0.05M PBS |
| Chemical Compound, Drug | Avidin | Vector Labs | Cat # PK-6106 RRID:AB_2336825 | [1:200] in 0.05M PBS |
| Chemical Compound, Drug | Biotin | Vector Labs | Cat # PK-6106 RRID:AB_2336825 | [1:200] in 0.05M PBS |
| Chemical Compound, Drug | NovaRED Perioxidase (HRP) Substrate Kit | Vector Labs | Cat # SK-4800 RRID: AB_2336845 | 18 drops (1), 12 drops (2), 12 drops (3) and 12 drops $H_2O_2$ solution in DI $H_2O$. |
| Chemical Compound, Drug | Triton | Sigma | Cat # T8787 | |
| Chemical Compound, Drug | Hydrogen Peroxide | Sigma | Cat # 216763 | |
| Chemical Compound, Drug | Paraformaldehyde | Sigma | Cat # P6148 | |
| Chemical Compound, Drug | Sucrose | Fisher Scientific | Cat # S5 | |
| Chemical Compound, Drug | Sodium Chloride | Fisher Scientific | Cat # S 640 | |
| Chemical Compound, Drug | Histoprep 100% Reagent Alcohol | Fisher Scientific | Cat # HC800 | |
| Chemical Compound, Drug | Histoprep 95% Reagent Alcohol | Fisher Scientific | Cat # HC1300 | |
| Chemical Compound, Drug | Histoclear II | Fisher Scientific | Cat # 5089990150 | |

*Continued on next page*

*Continued*

| Reagent type (species) or resource | Designation | Source or Reference | Identifier | Additional information |
|---|---|---|---|---|
| Chemical Compound, Drug | Omnimount | Fisher Scientific | Cat # 5089990146 | |
| Chemical Compound, Drug | 10% Neutral Buffered Formalin | Fisher Scientific | Cat # 22899402 | |
| Chemical Compound, Drug | Potassium Phosphate Monobasic | Fisher Scientific | Cat # P285 | |
| Chemical Compound, Drug | Potassium Phosphate Dibasic | Fisher Scientific | Cat # P288 | |
| Software and Algorithms | MED PC-IV | Med Associates | RRID:SCR_012156 | |
| Software and Algorithms | OmniPlex | Plexon | | Data Acquisition System |
| Software and Algorithms | Offline Sorter V6 | Plexon | RRID:SCR_000012 | |
| Software and Algorithms | NeuroExplorer | Plexon | RRID:SCR_001818 | |
| Software and Algorithms | Matlab | Mathworks | RRID:SCR_001622 | |
| Software and Algorithms | Statistica | StatSoft | RRID:SCR_014213 | |
| Software and Algorithms | SPSS | IBM | RRID:SCR_002865 | |
| Software and Algorithms | Adobe Illustrator | Adobe | RRID:SCR_010279 | |
| Software and Algorithms | Adobe Photoshop | Adobe | RRID:SCR_014199 | |
| Other | Plexon standard commutator | Plexon | Cat # 50122 | |
| Other | Plexon head stage cable | Plexon | Cat # 91809–017 | Metal Mesh Enclosed Cable |
| Other | Plexon head stage | Plexon | Cat # 40684–020 | |
| Other | Omnetics connector | Omnetics Corporation | Cat # A79042-001 | |
| Other | Green board - Moveable Array | San Francisco Circuits | Cat # PCB | |
| Other | Stainless Steel ground wire | AM Systems | Cat # 791400 | |
| Other | Formvar Insulated Nichrome Wire | AM Systems | Cat # 761500 | |
| Other | Dustless Precision Pellets | Bio-Serv | Cat # F0021 | |

## Subjects

Ten adult male rats at postnatal day 55 (P55) were obtained from Charles River Laboratories in Raleigh, NC. On arrival, rats were single-housed on a 12 hr light cycle (lights off at 6:00pm) and allowed three acclimation days with ad libitum access to water and standard chow (18% Protein Rodent Diet #2018, Harlan Teklad Global Diets, Madison, WI) prior to surgery. Rats were implanted with drivable, sixteen-wire microelectrode bundles. Each animal received between eleven and sixteen days to recover from surgery with ad libitum access to water and standard chow. Throughout the experiment, rats had ad libitum access to water; however, to generate motivation for a food-reward, standard chow was restricted to maintain rats at 85% of their free-feeding body weight. Three rats were eliminated from the study because electrodes failed to register single-unit activity

and one rat was eliminated due to incorrect electrode placement. Reported data are from remaining six individuals. All protocols were approved by the Boston College Animal Care and Use Committee and all experiments were carried out in accordance with the NIH guidelines regarding the care and use of rats for experimental procedures.

## Electrode assembly

Microelectrodes were constructed on site and consisted of a drivable bundle of sixteen Formvar-Insulated Nichrome wires (25.4 µm diameter: 761500, A-M Systems, Carlsborg, WA) within a 27-gauge cannula (B000FN3M7K, Amazon Supply). The cannula bundle was attached to a manually operated microdrive calibrated to permit ~0.042 mm advancement increments. Two free-hanging 127 µm diameter PFA-coated stainless-steel ground wires were also part of the assembly (791400, A-M Systems, Carlsborg, WA). All wires were electrically connected to a Nano Strip omnetics connector (A79042-001, Omnetics Connector Corp., Minneapolis, MN) on a custom 24-contact, individually-routed and gold-immersed circuit board (San Francisco Circuits, San Mateo, CA).

## Surgery

Aseptic stereotaxic surgery was performed under isoflurane anesthesia (1% to 5% in oxygen). Prior to incision, Rimadyl/Carprofen (024751, Henry Schein Animal Health, s.c. 5 mg/kg) and Ringer's lactate solution (014792, Henry Schein Animal Health, s.c. 2 to 5 mL) were administered sub-cutaneously to the back, and 2% lidocaine (002468, Henry Schein Animal Health, s.c. 0.25 mL) was administered sub-cutaneously above the skull. Post-incision, the skull was scoured in a crosshatch pattern with a scalpel blade to strengthen implant adhesion. Five screws (two anterior to Bregma, two between Bregma and lambda: 3 mm medial to the lateral ridges of the skull, and one on the midline: 5 mm posterior of lambda) were installed in the skull to further stabilize the bond between the skull, electrode assembly and protective head cap. A 1.4 mm diameter burr hole was drilled through the skull, centered on the implant site and the underlying dura was removed to expose the cortex. Nichrome recording wires were freshly cut with surgical scissors to extend approximately 2.0 mm beyond the cannula at a 15° angle. Just before implant, current was delivered to each recording wire in a saline bath, stripping each tip of its formvar insulation. Each omnetics connector contact was stimulated for 2 s using a resistor-equipped lead; current was supplied by a 12 V lantern battery. Machine grease was placed by the cannula and on the microdrive to prevent orthodontic resin from seizing moveable components.

The electrode assembly was slowly advanced at a 20° angle for implantation dorsal to the vlPAG. Coordinates from cortex: anterior-posterior (AP) −8.00 mm, medial-lateral (ML) −2.45 mm, and dorsal-ventral (DV) −5.52 mm. Once in place, stripped ends of both ground wires were wrapped around the posterior midline screw inserted previously. The microdrive base and a protective head cap surrounding the electrode assembly were cemented in place on the skull with orthodontic resin (C 22-05-98, Pearson Dental Supply, Sylmar, CA). At the end of the procedure, the omnetics connector was affixed to the head cap.

## Behavior apparatus

The apparatus for Pavlovian fear conditioning consisted of two individual behavior chambers with clear acrylic walls and top, and a grid floor with an acrylic waste pan below. Each grid floor bar was electrically connected to an aversive shock generator (Med Associates, St. Albans, VT) through a custom grounding device, which permitted the floor to be grounded at all times except during shock delivery. A nose poke opening equipped with infrared photocells was mounted on a central, acrylic wall panel and an acrylic external food cup was mounted on the same wall panel three inches below. Each behavior chamber was enclosed in a separate sound-attenuating shell. Auditory stimuli were presented through two speakers mounted on the ceiling of the shell, above the behavior chamber.

## Nose poke acquisition

On P56 and P57, all rats received 5 g of test pellets in their home cage. On P58, all rats received 30 test pellets released (one per minute) in the behavior chamber food cup (F0021, Bio-Serv, Flemington, NJ). On P59, all rats were shaped to nose poke for pellet delivery in the behavior chamber using a fixed ratio (FR1) schedule in which one nose poke yielded one pellet. Shaping sessions lasted 30

min or until approximately 50 nose pokes were completed. On P60, all rats received one variable interval (VI30) session in which nose pokes were reinforced on average every 30 s. On P61-P64 (inclusive) all rats received four variable interval (VI60) sessions in which nose pokes were reinforced on average every 60 s. For the remainder of behavioral testing, nose pokes were reinforced on a VI60 schedule independent of all Pavlovian contingencies.

### Pre-Exposure

On P65 and P66, all rats received one, 42 min session of pre-exposure to the three cues to be used in Pavlovian discrimination. Pre-exposure consisted of four presentations of each cue (12 total presentations) with mean inter-trial intervals (ITIs) of 3.5 min. The order of trial type presentation was randomly determined by the behavioral program and differed for each rat during each session. Auditory cues were 10 s in duration and consisted of repeating motifs of a broadband click, phaser, or trumpet (listen or download: http://mcdannaldlab.org/resources/ardbark).

### Fear discrimination

Prior to single-unit recording sessions, each rat received eight, 93 min sessions (one per day) of fear discrimination, consisting of 32 cue trials with mean ITIs of 3.5 min. Each 10 s auditory cue was associated with a unique probability of foot shock (0.5 mA, 0.5 s): danger, p = 1.00; uncertainty, p = 0.375; and safety, p = 0.00. Cue identity was counterbalanced across rats. Foot shock was administered 2 s following the termination of the auditory cue on danger and uncertainty shock trials. This was done in order to observe possible neural activity during the delay period not driven by an explicit cue. A single session consisted of six danger trials, ten uncertainty no-shock trials, six uncertainty shock trials, and ten safety trials. The order of trial type presentation was randomly determined by the behavioral program, and differed for each rat during each session. After the eighth discrimination session, rats were given ad libitum access to standard rat chow for at least 24 hr, followed by stereotaxic surgery. Following recovery, discrimination (identical to that described above) resumed with single-unit recording. Animals received discrimination with recording every other day. After each discrimination session with recording, electrodes were advanced either 0.042 mm or 0.084 mm to record from new units during the following session.

### Histology

Rats were deeply anesthetized using isoflurane and final electrode coordinates were marked by passing current from a 6 V battery through 4 of the 16 nichrome electrode wires. Rats were perfused with 0.9% biological saline and 4% paraformaldehyde in a 0.2 M potassium phosphate buffered solution. Brains were extracted and post-fixed in a 10% neutral-buffered formalin solution for 24 hr, and a 10% sucrose/formalin solution for an additional 24 hr before microtome sectioning. All brains were processed for light microscopy with immunohistochemistry for anti-tryptophan hydroxylase (T8575, Sigma-Aldrich, St. Louis, MO) coupled with a NovaRed chromagen reaction (SK-4800, Vector Laboratories, Burlingame, CA). Sections were mounted and imaged using a light microscope to confirm electrode placement.

### Single-unit data acquisition

During recording sessions, a 1x amplifying head stage connected the Omnetics connector to the commutator via a shielded recording cable (Head stage: 40684–020, Cable: 91809–017 and Commutator: 50122, Plexon Inc, Dallas TX). Analog neural activity was digitized and high-pass filtered via amplifier to remove low-frequency artifacts and sent to an Ominplex D acquisition system (Plexon Inc, Dallas TX). Behavioral events (cues, shocks, nose pokes) were controlled or recorded by a computer running Med PC-IV software (Med Associates, St. Albans, VT). Timestamped events from Med PC-IV were sent to the Ominplex D acquisition system via a dedicated interface module (DIG-716B). This acquisition process resulted in a single file (.pl2) containing all time stamps for all spikes and events. Single-units were sorted offline with a template-based spike-sorting algorithm (Offline Sorter V3, Plexon Inc, Dallas TX). Timestamped spikes and events (cues, shocks, nose pokes) were extracted and analyzed with statistical routines in MATLAB (Natick, MA). Neural activity was recorded throughout the 500 ms shock delivery period. Data are not presented from this period because we cannot be certain that shock artifacts did not disrupt spike collection.

## Statistical analysis

### Calculating suppression ratios

Fear was measured by suppression of rewarded nose poking, calculated as a ratio: (baseline poke rate – cue poke rate) / (baseline poke rate + cue poke rate). A ratio of '1.00' indicated high fear, '0.00' low fear, and gradations between reflect intermediate levels of fear.

### Behavior analyses

Behavior was analyzed using analysis of variance (ANOVA) with trial type as a factor. ANOVA for behavior contained three trial types (danger, uncertainty and safety). Uncertainty trial types (shock and no-shock) were collapsed because they did not differ for suppression ratio; during cue presentation, rats did not know the current uncertainty trial type. 95% bootstrap confidence intervals were constructed to evaluate the relationship between suppression ratios for each cue pair.

### 95% bootstrap confidence intervals

95% bootstrap confidence intervals were constructed using the bootci function in Matlab. For each bootstrap, a distribution was created by sampling the data 1000 times with replacement. Studentized confidence intervals were constructed with the final outputs being the mean, lower bound and upper bound of the 95% bootstrap confidence interval.

### Identifying cue-inhibited vlPAG neurons

All 245 neurons were screened for inhibitory firing during the first or last 5 s of danger and uncertainty cue presentation. This was achieved using a paired, two-tailed t-test comparing raw firing rate (spikes/s) during the 10 s baseline period just prior to cue onset with firing during the first or last, 5 s of cue presentation ($p < 0.0125$; Bonferroni corrected for six comparisons). Safety-responsive neurons were excluded because few neurons showed significant decreases in firing to safety.

### Z-Score normalization

For each neuron, and for each trial, firing rate (spikes/s) was calculated in 250 ms bins from 20 s prior to cue onset to 20 s following cue offset, for a total of 200 bins. Differential firing was calculated for each bin (n = 200) by subtracting mean baseline firing rate (2 s prior to cue onset) on that trial. Differential firing for each single-unit was Z-score normalized across all trials such that mean firing = 0, and standard deviation in firing = 1. Z-score normalization was applied to firing across all 200 bins, as opposed to only the bins prior to cue onset, in case neurons showed little/no baseline activity. Z-score normalized firing was analyzed with ANOVA using bin and trial-type as factors (*Figure 2A&C*). F and p values are reported, as well as partial eta squared and observed power.

### Identifying flip and sustain neurons

Normalized firing (Z-score) of each cue-inhibited neuron was averaged over the first (early) and last (late) 5 s of danger cue presentation. K-mean's clustering (k = 2) applied to early and late firing of all danger-inhibited neurons (n = 95) revealed two clusters of approximately equal size. Neuron identity screening at this stage revealed four neurons previously analyzed in a separate manuscript. These neurons were removed and did not undergo further analyses. This manuscript considers 91 cue-inhibited neurons for analysis: Flip neurons (n = 45), which were inhibited early but excited late, and Sustain neurons (n = 46), which maintained inhibition throughout danger cue presentation.

### Waveform analyses

Baseline firing rate, half duration and amplitude ratio of the mean waveform were determined for each Flip and Sustain neuron. Baseline firing rate (spikes/s) was calculated using the 10 s baseline period just prior to cue presentation. Half duration was determined by measuring the time (ms) from peak depolarization to the trough of after-hyperpolarization and dividing by 2. Amplitude ratio was calculated using $(n – p) / (n + p)$, where p = initial hyperpolarization (in mV) and n = maximal depolarization (in mV).

## Population firing analyses

Flip and Sustain population firing (*Figure 2*) were analyzed using analysis of variance (ANOVA) with trial type and bin (250 ms) as factors. ANOVA for normalized firing contained three trial types: danger, uncertainty and safety. Uncertainty trial types were collapsed because they did not differ for either suppression ratio or firing analysis. This was expected; during cue presentation rats did not know the current uncertainty trial type. F statistic, p value, observed power and partial eta squared are reported for effects and interactions. Bootstrap confidence intervals were performed for mean normalized firing to danger vs. uncertainty and uncertainty vs. safety during the first (early) and last (late) five seconds of cue presentation. Biases in single-unit firing to the three cues during the first and last 5 s (early or late) of cue presentation were determined using a sign test [p(s)] comparing normalized firing to danger vs. uncertainty and uncertainty vs. safety. The linear relationship between each cue firing comparison was determined using Pearson's correlation coefficient ($R^2$); associated p values for each comparison are also reported.

## Single-unit linear regression

Single-unit linear regression was used to determine the degree to which fear output and threat probability explained trial-by-trial variation in single-unit firing during specific 1 s cue intervals. The 32 trials composing a single session were ordered by trial type and Z-score normalized firing was specified for each trial and interval. The fear output regressor was the mean suppression ratio for the entire 10 s cue for the specific trial. The probability regressor was the foot shock probability associated with each cue (danger = 1.00, uncertainty = 0.375, safety = 0.00). The regression output of greatest interest is the beta coefficient (β) for each regressor (fear output and threat probability), which quantifies the strength (greater distance from zero = stronger) and direction (>0 = positive, <0 = negative) of the predictive relationship between each regressor and single-unit firing. ANOVA, bootstrap confidence intervals, sign test and Pearson's correlation coefficient were all used to analyze beta coefficients for Z-score normalized firing.

## Threat probability tuning curve

Nine separate regression analyses were performed as above. Only now, the value assigned to the uncertainty component of the threat probability regressor was systematically increased from 0 to 1 in 0.125 steps (0.000, 0.125, 0.250, 0.375, 0.500, 0.625, 0.750, 0.875 and 1.000). The first regression used the value of 0.000, second regression 0.125 and so on. Regression was performed for each 1 s interval of the 10 s cue. Beta coefficients for the first 5 s of cue and the last 5 s of cue were averaged to produce early and late threat tuning curves.

## Acknowledgements

We thank Dr. Hiram Brownell for statistical advice on linear regression, Dr. Donald B Katz for guidance on electrode construction, Dr. Thomas Stalnaker for guidance on waveform analysis, and Blake Zimmerman for guidance on Matlab scripts. Perceptually uniform colour-maps are used in this study to prevent visual distortion of the data (*Crameri, 2018a*; *Crameri, 2018b*).

## Additional information

### Funding

| Funder | Grant reference number | Author |
| --- | --- | --- |
| National Institutes of Health | R01MH117791 | Michael A McDannald |
| National Institutes of Health | R00DA034010 | Michael A McDannald |
| National Science Foundation | 5106201 | Kristina M Wright |

The funders had no role in study design, data collection and interpretation, or the decision to submit the work for publication.

## Author contributions
Kristina M Wright, Conceptualization, Data curation, Formal analysis, Visualization, Writing—original draft, Writing—review and editing; Thomas C Jhou, Resources, Writing—review and editing; Daniel Pimpinelli, Data curation, Constructed elctrodes used for recording, Made auditory devices used for cue presentation, Performed all post-operative care for rats, Assisted in training rats, Assisted with histology; Michael A McDannald, Conceptualization, Resources, Data curation, Software, Formal analysis, Supervision, Funding acquisition, Validation, Investigation, Visualization, Methodology, Writing—original draft, Project administration, Writing—review and editing

## Author ORCIDs
Kristina M Wright (iD) https://orcid.org/0000-0003-1446-3009
Thomas C Jhou (iD) http://orcid.org/0000-0001-8811-0156
Michael A McDannald (iD) https://orcid.org/0000-0001-8525-1260

## Ethics
Animal experimentation: This study was performed in strict accordance with the recommendations in the Guide for the Care and Use of Laboratory Animals of the National Institutes of Health. All of the animals were handled according to approved institutional animal care and use committee (IACUC) protocols (#2018-002) of Boston College. The protocol was approved by the Institutional Animal Care and Use Committee of Boston College. All surgery was performed under isoflurane anesthesia, and every effort was made to minimize suffering.

## Decision letter and Author response
Decision letter https://doi.org/10.7554/eLife.50054.012
Author response https://doi.org/10.7554/eLife.50054.013

# Additional files

## Supplementary files
• Transparent reporting form DOI: https://doi.org/10.7554/eLife.50054.007

## Data availability
Data have been deposited at: http://crcns.org/data-sets/bst/pag-1.

The following previously published dataset was used:

| Author(s) | Year | Dataset title | Dataset URL | Database and Identifier |
|---|---|---|---|---|
| Wright KM, McDannald MA | 2019 | Ventrolateral periaqueductal gray single-single unit activity during multi-cue fear discrimination in rats | http://crcns.org/data-sets/bst/pag-1 | Collaborative Research in Computational Neuroscience, PAG-1 |

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
