## [Decision Letter]

Thank you for submitting your article "Cue-inhibited ventrolateral periaqueductal gray neurons signal fear output and threat probability" for consideration by *eLife*. Your article has been reviewed by two peer reviewers, and the evaluation has been overseen by a Reviewing Editor and Laura Colgin as the Senior Editor. The following individuals involved in review of your submission have agreed to reveal their identity: Laura Bradfield (Reviewer #1); Satoshi Ikemoto (Reviewer #2).

The reviewers have discussed the reviews with one another and the Reviewing Editor has drafted this decision to help you prepare a revised submission.

Summary:

This manuscript follows on from prior work showing that vlPAG neurons do not simply signal fear output, as is the dominant current theory of vlPAG function, but that they better reflect threat probability. The current manuscript builds on these findings in a detailed and elegant manner, demonstrating that the firing patterns of neuronal populations within vlPAG are in fact, heterogeneous, and that a subpopulation of neurons show early inhibition to a danger cue, but that this response becomes excitation throughout the 10 s of cue presentation. This pattern of firing was concluded to reflect threat timing. By contrast, an second population of about the same size, sustained their inhibitory firing throughout the cue signal, and this firing was greatest for the danger signal, then for the uncertainty signal, and was not present for the safety signal. The pattern of firing for this population was revealed to better reflect fear output than threat probability, and the implications of this for current theories of fear conditioning are discussed.

Overall the reviewers found this article to be excellent. The findings are novel and exciting, the methodologies employed were solid, and the discussion of the results was thoughtful and insightful. There was only one more major comment regarding statistical analysis that we would like the authors to address in a revised version of the manuscript.

Essential revisions:

The reviewers had a concern regarding the statistical analyses. Specifically, the authors sometimes violate statistical conventions in analyzing the data. If the authors choose to violate conventions, they must justify them.

Results, second paragraph and subsection “Flip and Sustain populations show differential cue firing”: Conduct posthoc tests, instead of t-tests, after a significant main effect of the ANOVAs on trial type. t-Tests do not account for family-wise errors.

Figure 4F legend: Bonferroni correction was performed for 14 t-tests. However, for Figure 4A, it should be done in 28 t-tests, which include both regressors. For Figure 4D, t-tests should not be performed separately between the regressors for each 1-s bin, because the ANOVA indicates just a main significant effect on interval without interaction between interval and regressor. Having that said, there appear to be difference between the regressors during the post cue period, suggesting that insufficient power to detect such interaction with the ANOVA. The authors may want to consider performing three ANOVAs: one for baseline, one during the cue, and one after cue.

---

## [Author Response]

Essential revisions:The reviewers had a concern regarding the statistical analyses. Specifically, the authors sometimes violate statistical conventions in analyzing the data. If the authors choose to violate conventions, they must justify them.Results, second paragraph and subsection “Flip and Sustain populations show differential cue firing”: Conduct posthoc tests, instead of t-tests, after a significant main effect of the ANOVAs on trial type. t-Tests do not account for family-wise errors.

We agree, and hope you find the adjustments we have made suitable in addressing this concern. As data were analyzed using SPSS, posthoc options were not available due to the lack of between subject comparisons. In place of all t-tests, we now use bootstrapping to construct 95% confidence intervals for each of the instances mentioned above. Confidence intervals allow us to make between-cue comparisons without assuming a normal distribution and also help circumvent the problem of multiple comparisons.

Figure 4F legend: Bonferroni correction was performed for 14 t-tests. However, for Figure 4A, it should be done in 28 t-tests, which include both regressors. For Figure 4D, t-tests should not be performed separately between the regressors for each 1-s bin, because the ANOVA indicates just a main significant effect on interval without interaction between interval and regressor. Having that said, there appear to be difference between the regressors during the post cue period, suggesting that insufficient power to detect such interaction with the ANOVA. The authors may want to consider performing three ANOVAs: one for baseline, one during the cue, and one after cue.

Thank you for pointing this out, in place of t-tests, we now use bootstrapping and construct 95% confidence intervals to indicate where β coefficients for each regressor differ from zero. We feel this provides a clearer demonstration of the overall pattern, which is supported by the scatter plots in Figure 4. As suggested, we have adjusted our ANOVA approach and performed three ANOVAs for baseline, cue and post-cue periods. Due to this new approach, we are able to detect the post-cue main effect of regressor for Sustain neurons.